# Research on Steel Surface Defect Detection Based on YOLOv5 with Attention Mechanism

**Jianting Shi** [1,*]**, Jian Yang** [2]🆔 **and Yingtao Zhang** [3]🆔

1 School of Computer and Information Engineering, Heilongjiang University of Science and Technology, Harbin 150022, China
2 Graduate School, Heilongjiang University of Science and Technology, Harbin 150022, China
3 School of Computer Science and Technology, Harbin Institute of Technology, Harbin 150001, China
* Correspondence: hotmail8194@163.com

**Abstract:** Due to the irresistible factors of material properties and processing technology in the steel production, there may be different types of defects on the steel surface, such as rolling scale, patches and so on, which seriously affect the quality of steel, and thus have a negative impact on the economic efficiency of the enterprises. Different from the general target detection tasks, the defect detection tasks have small targets and extreme aspect ratio targets. The contradiction of high positioning accuracy for targets and their inconspicuous features makes the defect detection tasks difficult. Therefore, the original YOLOv5 algorithm was improved in this paper to enhance the accuracy and efficiency of detecting defects on steel surfaces. Firstly, an attention mechanism module was added in the process of transmitting the shallow feature map from the backbone structure to the neck structure, aiming at improving the algorithm attention to small targets information in the feature map and suppressing the influence of irrelevant information on the algorithm, so as to improve the detection accuracy of the algorithm for small targets. Secondly, in order to improve the algorithm effectiveness in detecting extreme aspect ratio targets, K-means algorithm was used to cluster and analyze the marked steel surface defect dataset, so that the anchor boxes can be adapted to all types of sizes, especially for extreme aspect ratio defects. The experimental results showed that the improved algorithms were better than the original YOLOv5 algorithm in terms of the average precision and the mean average precision. The mean average precision, demonstrating the largest increase among the improved YOLOv5 algorithms, was increased by 4.57% in the YOLOv5+CBAM algorithm. In particular, the YOLOv5+CBAM algorithm had a significant increase in the average precision for small targets and extreme aspect ratio targets. Therefore, the YOLOv5+CBAM algorithm could make the accurate localization and classification of steel surface defects, which can provide a reference for the automatic detection of steel defects.

**Keywords:** steel surface defects; small targets; attention mechanism; K-means



## 1. Introduction

As the level of industrial production continues to improve, enterprises are more and more seriously concerned about steel quality, and it is crucial to have qualified steel with higher criteria, especially in some high-tech industries, such as automobile, aerospace, machinery and electronics. Due to the influence of various irresistible factors in the steel production, there are a variety of defects on the steel surface, such as rolling scale, patches and so on. These defects affect the toughness and wear resistance of steel, causing huge economic losses to enterprises. Therefore, the steel industry attaches great importance to the detection of steel surface defects [1–3].

With the development of machine vision [4–6], many scholars have studied the detection algorithms of steel surface defects. Suvdaa et al. [7] proposed an algorithm based on scale-invariant feature transform (SIFT) and support vector machine (SVM), then applied it

to steel plate surface defect detection. Song et al. [8] improved the complete local binary pattern (CLBP) around the adjacent evaluation window. Jeon et al. [9] used a dual illumination structure to distinguish the color change caused by uneven defects and surface noise and used Gabor filter and binarization method to extract the shape of defects. Gyimah et al. [10] used a combination of non-local (NL) means to filter with wavelet thresholding and CLBP to extract robust features, which were fed into classifiers for surface defects detection. The above studies on steel defects only classified the defect types, and the adopted methods were to extract image features using traditional methods, such as CLBP and SIFT, then used machine learning techniques to classify the features. Although the speeds of these methods are fast in detection and the classification accuracy is high for specific defects, traditional feature extraction methods are not as effective in extracting features for defects with inconspicuous textures. Additionally, most of the early work remained at the stage of steel defect classification. As the improvement of industrial technologies and the complexity of manufacturing steel, various defects on steel surface are veiled under cameras. Thus, the only classification of steel defects cannot meet the increasing industrial demand.

In recent years, deep learning has gradually emerged and achieved excellent results in many industries. In the tasks related to image processing, deep learning can extract more abundant features of images, compared with traditional feature extraction methods. Deep learning technology provides a new idea for steel surface defect detection, and it is of great significance to apply deep learning to steel enterprises [11,12]. YOLO algorithms are the one-stage target detection algorithm [13] widely used in industry, which take both speed and accuracy into account. Faster R-CNN algorithm [14] is representative of the two-stage target detection algorithm because it divides the generation of candidate boxes and prediction into two stages, its detection accuracy is high. Hatab et al. [15] detected defects in the NEU-DET dataset, mainly by modifying the hyperparameters of the YOLOv3 algorithm, i.e., the size of the input and the size of the batch. Since no specific improvements were made to the algorithm according to the characteristics of the defects, the mean average precision of the algorithm is as low as 70.66%. Furthermore, the algorithm was less effective in detecting small targets, for example, the average precisions of rolled-in scale and inclusion with a large number of small targets were only 62.31% and 72.05%. Kou et al. [16] improved YOLOv3 algorithm to detect defects in the NEU-DET dataset. The anchor-free [17] feature selection mechanism was utilized to select an ideal feature scale for model training, and specially designed dense convolution blocks were introduced into the model to extract rich feature information. This improved algorithm had a low mean average precision and anchor-free caused the number of positive and negative samples to become unbalanced, which led to recalls and precisions polarization. Additionally, the algorithm still had poor detection ability for small targets because the algorithm was not improved according to the characteristics of defects. Wei et al. [18] investigated four types of defects (scratch, indentation, crust, and fold). Relatively speaking, indentation defects and crust defects had mainly widths and heights in the range of 60~80 pixels, so they were small targets. Scratch defects and fold defects had mainly heights of 200 pixels and widths of 60 pixels, so they were extreme aspect ratio targets. Based on the Faster R-CNN algorithm, rol (weighted region of interest pooling) was introduced to solve the problem of missing detection of a large number of small targets, then deformable convolution [19] and feature pyramid were used to deal with the irregularity and diversity of defects. Although this method had high detection rate in actual industrial environment, only four types of defects were detected, and the deformable convolution increased the computation of the algorithm, resulting in a lower detection speed. Ning et al. [20] detected defects in the NEU-DET dataset. A layer of prediction box was added to YOLOv3 algorithm to improve the detection ability of the algorithm for small defects, and K-means++ algorithm was used to cluster defect labels. Compared with the original YOLOv3 algorithm, the mean average precision of the improved algorithm increased by 14.7%. However, due to the addition of a layer of prediction box, the algorithm weight file became larger, resulting in a slower detection speed of 34FPS for the algorithm, which was far from meeting the

real-time detection requirements of enterprises. M Li et al. [21] detected defects in the NEU-DET dataset. An attention mechanism was added to the backbone structure of the YOLOv4 algorithm, and the path aggregation network was modified into a customized receptive field block structure. The mean average precision of improved algorithm in the detection was improved by 3.87%, reaching 85.41%. However, only four types of defects (inclusion, patches, pitted_surface and scratches) were detected, and the algorithm detection speed became slow due to the massive improvements made to the backbone structure. In addition, this method is less effective in detecting small targets, for example, the average precision of inclusion with a large number of small targets was only 74.1%. Zeqiang et al. [22] detected defects in the NEU-DET dataset. An attention mechanism was added to the feature extraction network, and the filtered weighted feature vector was used to replace the original feature vector for residual fusion. Then, the convolution layer was added after the pooling structure of spatial pyramid in order to improve the ability of defect feature extraction. Although the mean average precision of this algorithm was high, the detection speed of the algorithm was low due to a large number of improvements to the FPN structure. Additionally, because the feature maps lost a lot of feature information after pooling, adding an attention mechanism after the shallow feature map was pooled will result in the attention mechanism not getting enough location information, leading to the low average precisions of the algorithm for small targets.

In this paper, the size characteristics of steel surface defects are discussed. Firstly, for steel surface defects with more small targets, an attention mechanism module is used to enhance the attention of the algorithm to the information of the small targets. Secondly, for the extreme aspect ratio targets, the K-means algorithm is used to cluster the anchor boxes. The experiments show that the proposed algorithm in this paper can improve the detection ability of small targets and extreme aspect ratio targets, achieving the classification and localization of steel surface defects with higher accuracy and efficiency than the ordinary steel surface defect algorithm.

## 2. Methods

### 2.1. Dataset Analysis

The number of steel images is, in total, 1166, and the heights and widths of the images are 200 pixels. The experiments are conducted using Pascal VOC 2007 dataset format, which is authoritative in the fields of image detection, classification recognition and semantic segmentation. The training set and validation set are divided with the test set in the ratio of 9:1, and then the training set is divided with the validation set in the ratio of 9:1. The division results in 944 images for the training set, 105 images for the validation set and 117 images for the test set.

Steel surface defects are classified into a total of 33 types based on international uniform regulations. In this paper, five common types of steel surface defects are selected based on the Northeastern University steel surface defect dataset (NEU-CLS) and international steel surface defect types. Furthermore, the names of defect types in the dataset are abbreviated in this paper, i.e., RS is for rolled-in scale, PA is for patches, IN is for inclusion, PS is for pitted_surface and SC is for scratches, in order to further facilitate the proper names. The images of five common defects on the steel surface are shown in Figure 1, and according to the image labels of the dataset, the ground truth boxes of defects are marked with green rectangular boxes.

RS defects are black spots in the form of dots, strips or fish scales, which have a small impact on the quality of steel, but due to this, the features are not obvious, causing a negative impact on the detection of defects on the steel surface. PA defects are irregular shapes, black and gray in color, which have a negative impact on the quality and appearance of steel. IN defects are non-metallic inclusions in the shape of dots, blocks or lines, which mainly have a negative impact on the appearance of steel. PS defects are regional concave black spots with large or small areas, mainly caused by the poor quality of raw materials used in the production of steel or the high temperature of the production environment.

SC defects are straight and thin bright streaks, which are mostly caused by problems with mechanical equipment.

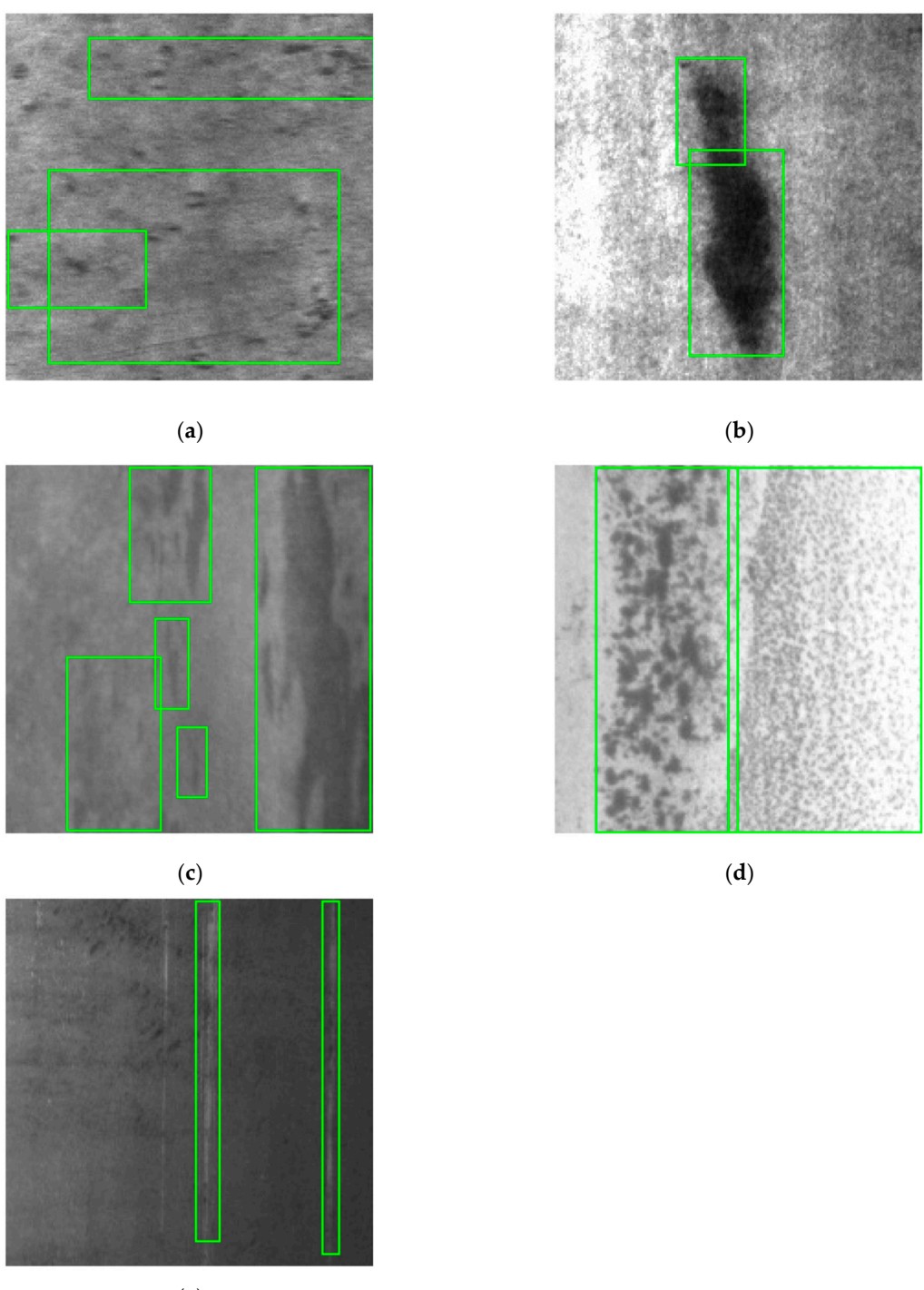

**Figure 1.** Five types of steel surface defects. (**a**) This is an image of the RS defect and the location of the defect is marked with a green rectangle; (**b**) This is an image of the PA defect and the location of the defect is marked with a green rectangle; (**c**) This is an image of the IN defect and the location of the defect is marked with a green rectangle; (**d**) This is an image of the PS defect and the location of the defect is marked with a green rectangle; (**e**) This is an image of the SC defect and the location of the defect is marked with a green rectangle.

Most of the usual target detections are to detect targets in life; small targets often have obvious features, which are distinct to humans, and there are almost no extreme aspect ratio targets. The defect detection task is to detect defects in industrial production process. Since small targets often do not have obvious features, they cannot be spotted by human eyes, let alone the extreme aspect ratio targets. In this paper, targets whose width and height both belong to less than 80 pixels are defined as small targets. Among the two values of width and height of a target, one of which is larger than 150 pixels and the other is smaller than 100 pixels, the target is defined as an extreme aspect ratio target.

In this paper, the scatter distribution statistics of the sizes of all types of defects are performed in the form of scatter plots. The statistical results of all types of defects are shown in Figure 2.

In Figure 2a, the widths and heights of RS defects are mainly concentrated in the range of 20~80 pixels, and the number of defects gradually decreases as the pixel value becomes larger. From this, the RS defects have a large number of small targets.

In Figure 2b, the widths of PA defects are mainly concentrated in the range of 20~80 pixels. However, the heights of PA defects span a wide range, and some defects are in the 200 pixel range. From this, the PA defects have a large number of small targets and extreme aspect ratio targets.

In Figure 2c, the widths of IN defects are mainly concentrated in the range of 20~50 pixels, and the heights of IN defects span a wide range. From this, the IN defects have a large number of small targets and extreme aspect ratio targets.

In Figure 2d, the heights of PS defects are large and are mainly concentrated in the range of 180~200 pixels. The widths of PS defects span a wide range and are evenly distributed in each range. From this, the PS defects have a large number of extreme aspect ratio targets.

In Figure 2e, for defects whose widths are concentrated in the range of 0~50 pixel, the heights are concentrated in the range of 50~200 pixels, and a large portion of them are in the range of 150~200 pixels. For defects whose heights are concentrated in the range of 0~50 pixels, the widths are concentrated in the range of 50~200 pixels, and the number of defects increases as the width increases. From this, the SC defects have a large number of extreme aspect ratio targets.

Overall, the defects have various sizes, mainly small targets and extreme aspect ratio targets, which put high demands on the target detection algorithm. The algorithm should not only improve the detection accuracy of small and extreme aspect ratio targets but also improve the detection effectiveness for the overall defects.

*2.2. YOLOv5*

YOLO algorithms [23–25] are one-stage target detection algorithms. Compared with other algorithms, they have better performances under the same size and have steadily improved. Introduced by Glenn Jocher in 2020, the YOLOv5 algorithm has greatly improved both detection speed and detection accuracy. There are four versions of YOLOv5: 5 s, 5 m, 5 L and 5 x. In this paper, the YOLOv5l algorithm will be improved to enhance the accuracy of steel surface defect detection. YOLOv5 [26] algorithm structure includes input, backbone, neck and head. The algorithm structure of YOLOv5l is shown in Figure 3.

The input structure of YOLOv5 algorithm uses Mosaic data enhancement. Mosaic data enhancement uses four pictures, which are clipped, scaled, rotated and then synthesized into one image. Random clipping and stitching greatly enrich the dataset. This method is very effective for datasets with fewer images.

The feature extraction network of the YOLOv5 algorithm is the CSPDarkNet network, which mainly includes the residual, CSPNet, focus and SPP structures, as shown on the right in Figure 3. The advantage of the residual network [27] is that it is easy to optimize and improve the accuracy by increasing the depth of the network. There are two CSPNet structures [28], CSP1_X structure by backbone and CSP2_X structure by neck. The focus structure is used to slice and stitch the high-resolution feature maps into multiple low-

resolution feature maps. The SPP structure [29] serves to convert a feature map of arbitrary size into a fixed-size feature vector.

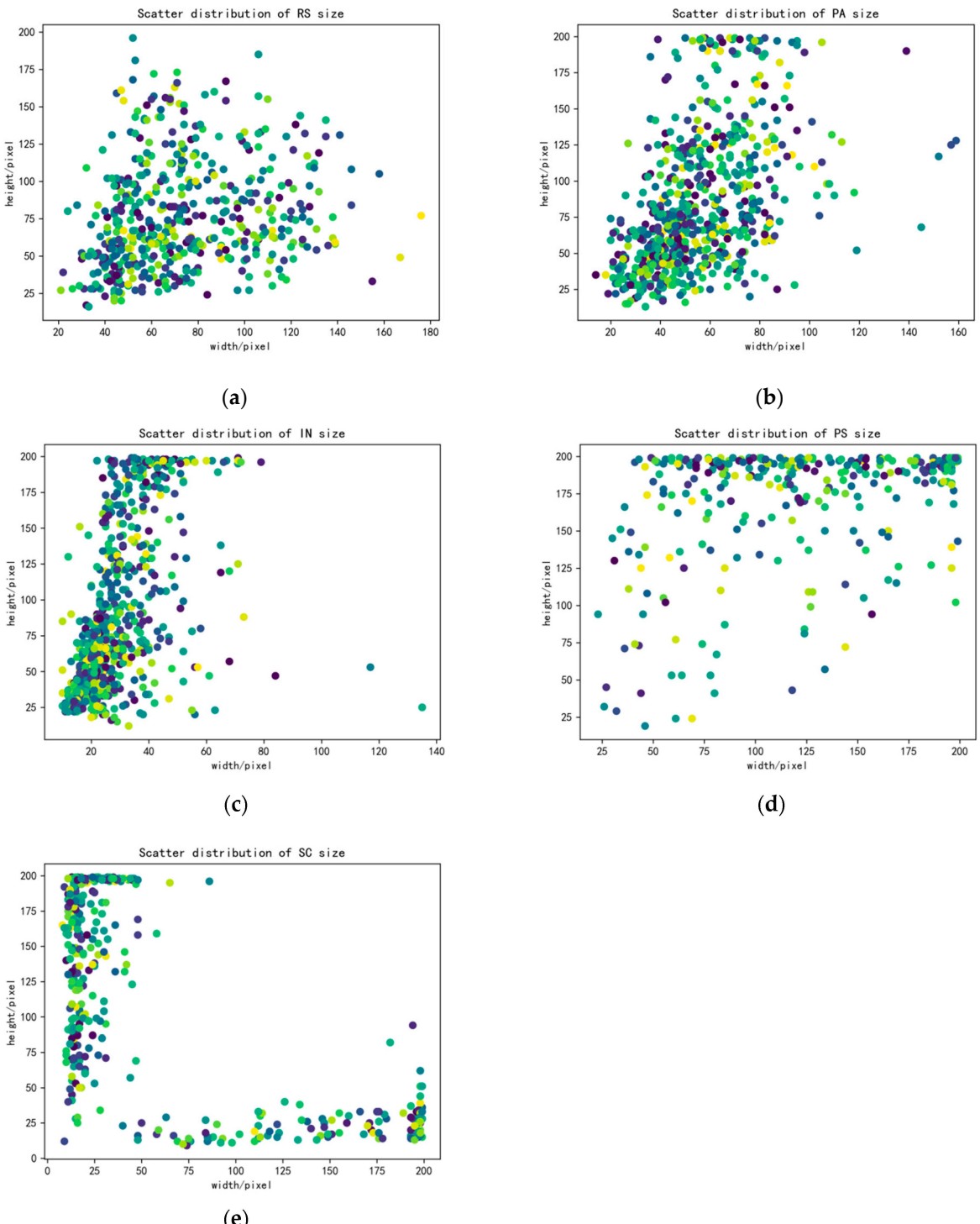

**Figure 2.** Size scatter distribution plots of all types of defects. (**a**) Size scatter distribution plot of RS defects; (**b**) Size scatter distribution plot of PA defects; (**c**) Size scatter distribution plot of IN defects; (**d**) Size scatter distribution plot of PS defects; (**e**) Size scatter distribution plot of SC defects.

The neck structure of the YOLOv5 algorithm combines the FPN structure [30] and the PAN structure [31]. The FPN structure transmits semantic information and the PAN structure transmits location information, enabling the network to fuse more feature information.

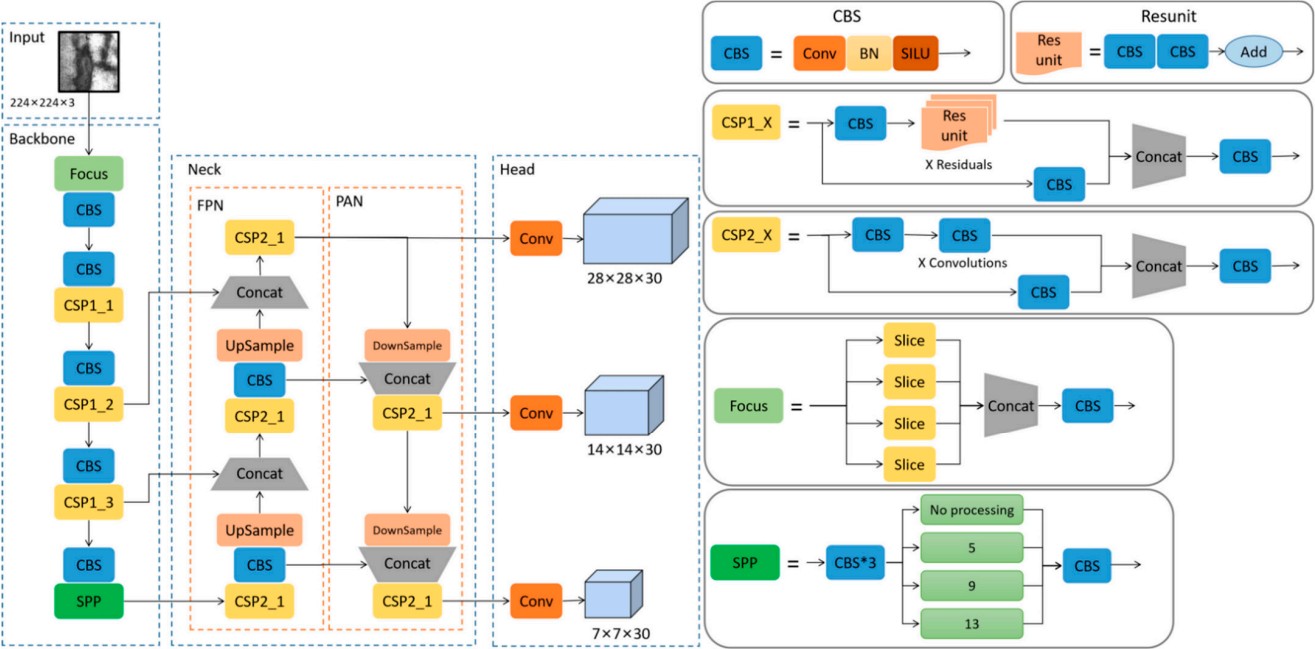

**Figure 3.** The structure diagram of Yolov5l algorithm.

The head structure of the YOLOv5 algorithm contains a convolution to adjust the number of channels of the feature map. YOLOv5 algorithm uses binary cross-entropy to calculate classification loss and confidence loss, and CIoU to calculate the localization loss. NMS is used to select the candidate box with the highest predicted probability as the final predicted box.

### 2.3. Evaluating Indicator

In the field of target detection, the most common type of evaluation metric is the mean average precision (mAP). In addition, an algorithm is evaluated by precision (P), recall (R), the average precision (AP) and F1-score.

Intersection over union (IoU) is the ratio between the intersection and the concatenation of the prediction box and the ground truth box, with the range 0–1. The IoU in this paper is 0.3, i.e., when the IoU is higher than 0.3, it is considered a valid detection, otherwise it is invalid.

Confidence indicates that the prediction box is a certain type of probability, with the range 0–1. When the confidence is higher than the confidence threshold, the prediction box is a positive sample, otherwise it is negative. In this paper, the confidence threshold is 0.5.

Precision is related to the prediction result, which indicates how many assumed positive samples are real positive ones. The lower the precision is, the more likely the algorithm is to false detection. The precision is calculated using Equation (1).

$$P = \frac{TP}{TP + FP} \tag{1}$$

Recall indicates the probability of a positive sample being predicted correctly. The lower the recall is, the more likely the algorithm is to miss detection. The recall is calculated using Equation (2).

$$R = \frac{TP}{TP + FN} \tag{2}$$

The average precision indicates the area of the region enclosed by the precision–recall curve and the coordinate axis. The average precision is calculated using Equation (3).

$$AP = \int_0^1 P(R)dR \tag{3}$$

The F1-score is the weighted summed average of precision and recall, and the weight is often 1, i.e., precision and recall are considered equally important. This is what reflects the strength of the algorithm to some extent. The *F1*-score is calculated using Equation (4).

$$F1 = \frac{2 * P * R}{P + R} \tag{4}$$

TP refers to predicting positive samples as positive. FP refers to predicting negative samples as positive. FN refers to predicting positive samples as negative. TN refers to predicting negative samples as negative.

## 3. Steel Defect Detection Algorithm Based on Improved YOLOv5

The attention mechanism [32] is an efficient module in the field of neural networks. It has been widely used in different directions, such as target detection, natural language processing and semantics segmentation, with good detection results. In the training process, the algorithms often need to accept and process all features in the images. However, algorithms only need to extract the features in the image only related to the targets and ignore the irrelevant features. The attention mechanism enables the algorithm to learn the ability to distinguish targets and backgrounds by training, selectively ignoring the unimportant areas of the image and fortifying the focus on the targeted ones.

### 3.1. SENet

The SENet [33] attention mechanism is a channel attention mechanism (CAM), which mainly consists of squeeze and excitation. SENet pays more attention to the differences between different channels of the feature maps and adjusts the attention weights according to these differences. The structure diagram of SENet is shown in Figure 4.

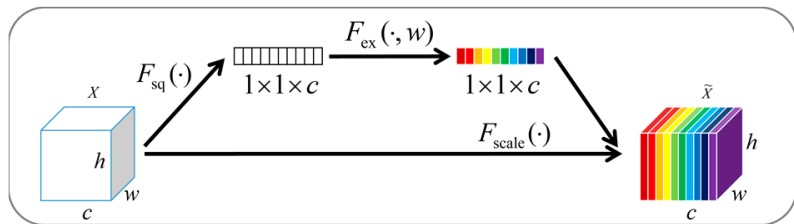

**Figure 4.** The structure diagram of SENet.

In Figure 4, $X$ denotes the feature maps of $h \times w$ with channel number $c$, and $X_c$ denotes the c-th feature map in $X$. $F_{sq}$ and $F_{ex}$ denote the squeeze operation and excitation operation, respectively.

### 3.2. CBAM

The CBAM [34] attention mechanism combines channel attention mechanism and spatial attention mechanism (SAM). CBAM fuses the attention weights in both channels and spatial dimensions on the basis of the input feature maps and multiplies them with the input feature map to obtain a new feature map, respectively, which is beneficial for extracting information on the feature map. The structure diagram of CBAM is shown in Figure 5.

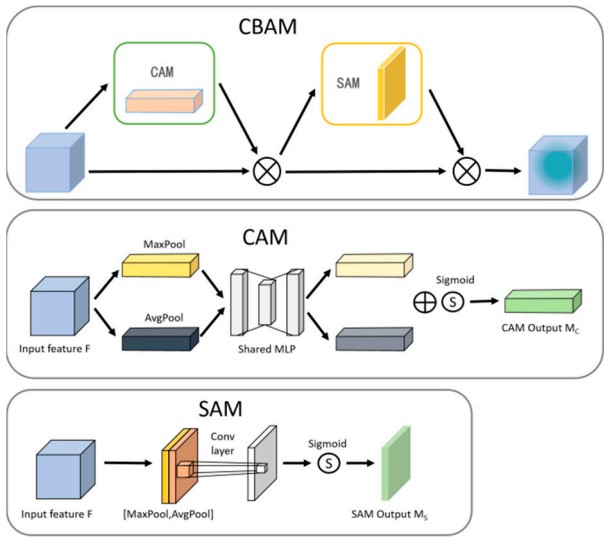

**Figure 5.** The structure diagram of CBAM.

### 3.3. ECANet

The ECANet [35] attention mechanism is a channel attention mechanism, which can be regarded as an improved version of SENet. SENet attention mechanism requires capturing the dependencies of all feature points of different channels, which greatly reduces the efficiency of extracting information of the targets. The structure diagram of ECANet is shown in Figure 6.

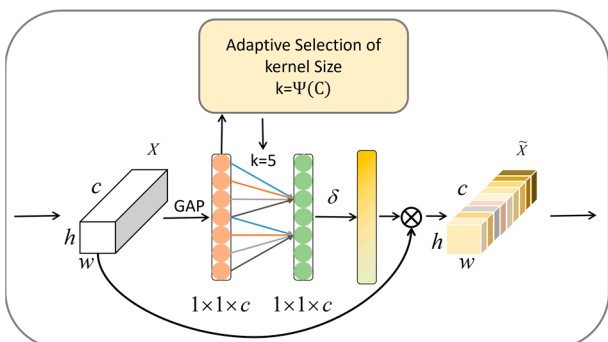

**Figure 6.** The structure diagram of ECANet.

### 3.4. Experimental Results and Analysis

In convolutional neural networks, the deeper the network layers are, the more semantic information the feature maps contain, but lesser location information means the information about small targets is easily lost in the process of continuous convolution. Therefore, the deep feature map contains more semantic information and is suitable for large targets detection, while the shallow feature map contains more location information and is suitable for small targets detection.

To verify the effect of the attention mechanism module on algorithm detection accuracy, three attention modules, SENet, CBAM and ECANet, are introduced into YOLOv5 algorithm. The attention mechanism module is a plug-and-play structure that can be placed in the backbone structure or in the neck structure. In this paper, one attention mechanism module is added to enhance the attention of the algorithm to the information of small targets.

In the process of extracting target feature by the backbone structure, the number of channels of the shallow feature map obtained by the second CSP structure is 256. In the process of transmitting the shallow feature map to the FPN structure, an attention

mechanism module is added to connect with the concat structure of the FPN structure, at which time the number of channels of the shallow feature map is expanded to 512.

The improved YOLOv5 algorithm is shown in Figure 7, in which CA is the attention mechanism.

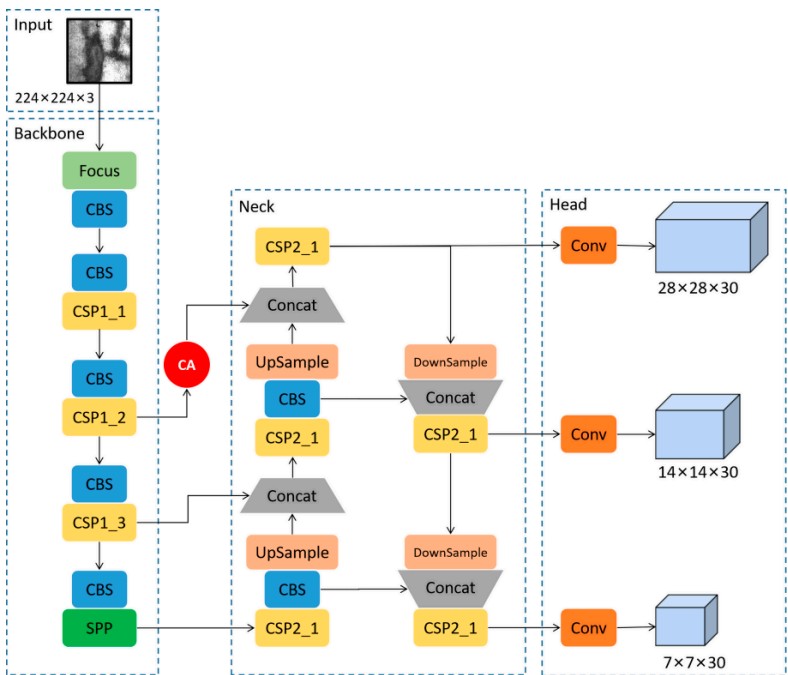

**Figure 7.** The structure diagram of YOLOv5 with the attention mechanism.

The sizes of anchor boxes used in the original YOLOv5 algorithm are not random values but are obtained by clustering according to the sizes of targets of the COCO dataset image. The COCO dataset is a general dataset, most of its images come from real-life scenes. The sizes of the ground truth boxes of the COCO dataset are quite different from the sizes of images of steel surface defect dataset.

In this paper, K-means [36] algorithm is used to cluster and analyze the defect dataset of marked steel surface, and nine anchor boxes are generated for the three feature maps of different sizes and are allocated to the three feature maps according to their sizes.

The clustering result obtained using the K-means algorithm is shown in Figure 8. The information of the feature maps and anchor boxes after clustering is shown in Table 1.

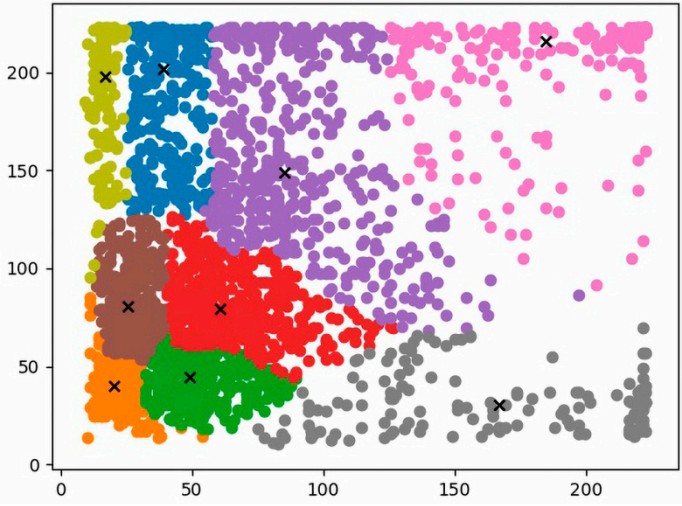

**Figure 8.** Clustering result.

**Table 1.** Anchor box information.

| Feature Map | Anchor Box Size |
|---|---|
| 28 × 28 | (20 × 40); (25 × 80); (49 × 44) |
| 14 × 14 | (16 × 198); (60 × 79); (166 × 30) |
| 7 × 7 | (39 × 201); (85 × 148); (184 × 216) |

The improved YOLOv5 algorithms are experimentally compared with YOLOv5 algorithm, YOLOv4 algorithm and Faster R-CNN algorithm. The algorithms are compared in multiple ways according to P, R, AP, mAP and F1-score; the experimental results are shown in Table 2.

**Table 2.** Comparison of detection effect of the improved algorithms, YOLOv4 algorithm, YOLOv5 algorithm and Faster R-CNN algorithm.

| Algorithm | Type | P | R | F1-Score | AP | mAP |
|---|---|---|---|---|---|---|
| YOLOv5 | RS | 80.95% | 36.17% | 0.5 | 62.09% | |
| | PS | 83.87% | 63.41% | 0.72 | 81.27% | |
| | IN | 92.31% | 58.82% | 0.72 | 79.65% | 81.78% |
| | PA | 93.18% | 82% | 0.87 | 93.23% | |
| | SC | 88.57% | 79.49% | 0.84 | 92.65% | |
| YOLOv5+SENet | RS | 83.87% | 55.32% | 0.67 | 71.7% | |
| | PS | 93.55% | 70.73% | 0.81 | 87.68% | |
| | IN | 90.41% | 64.71% | 0.75 | 82.76% | 85.83% |
| | PA | 91.3% | 84% | 0.87 | 92.24% | |
| | SC | 94.12% | 82.05% | 0.88 | 94.76% | |
| YOLOv5+CBAM | RS | 73.53% | 53.19% | 0.62 | 70.19% | |
| | PS | 96.55% | 68.29% | 0.8 | 87.85% | |
| | IN | 91.43% | 62.75% | 0.74 | 83.66% | 86.35% |
| | PA | 97.67% | 84% | 0.9 | 94.16% | |
| | SC | 92.31% | 92.31% | 0.92 | 95.91% | |
| YOLOv5+ECANet | RS | 79.41% | 57.45% | 0.67 | 71.01% | |
| | PS | 90.91% | 73.17% | 0.81 | 84.17% | |
| | IN | 89.86% | 60.78% | 0.73 | 81.26% | 84.61% |
| | PA | 95.56% | 86% | 0.91 | 92.64% | |
| | SC | 92.31% | 92.31% | 0.92 | 93.95% | |
| YOLOv4 | RS | 0% | 0% | 0 | 15.62% | |
| | PS | 82.61% | 46.34% | 0.59 | 70.63% | |
| | IN | 90.48% | 18.63% | 0.31 | 52.49% | 59.46% |
| | PA | 97.14% | 68% | 0.8 | 85.34% | |
| | SC | 89.47% | 43.59% | 0.59 | 73.24% | |
| Faster R-CNN | RS | 19.33% | 90.62% | 0.32 | 45.46% | |
| | PS | 47.37% | 79.41% | 0.59 | 79.98% | |
| | IN | 35.35% | 89.41% | 0.51 | 70.76% | 74.85% |
| | PA | 52.42% | 94.2% | 0.67 | 87.98% | |
| | SC | 57.97% | 90.91% | 0.71 | 90.09% | |

Compared with the original YOLOv5 algorithm, the mean average precisions of the algorithms introducing SENet, CBAM and ECANet increase by 4.05%, 4.57%, and 2.83%, respectively. Among them, YOLOv5+SENet and YOLOv5+ECANet perform better with their increased average precisions of RS, PS, IN, SC, and 0.99% and 0.59% decreases in the average precision of PA, respectively, with a small decrease. YOLOv5+CBAM performs the best with its increased average precisions of all types of defects, and its mean average precision reaches 86.35%, which is the highest.

In terms of the F1-score, the F1-scores of YOLOv5 algorithm for all types of defects are the worst. YOLOv5+SENet has the best F1-scores for RS, PS and IN. YOLOv5+CBAM has the best F1-score for SC. YOLOv5+ECANet has the best F1-scores for RS, PS, PA and SC. This shows that the improved algorithms are better than the original YOLOv5 algorithm to a certain extent.

For the steel enterprises, it is more important to avoid miss-selection than to avoid false inspection in the real manufacture process [30]. In terms of the recall, the recalls of YOLOv5 algorithm for all types of defects are the worst. YOLOv5+SENet has the best recall for IN. YOLOv5+CBAM has the best recall for SC. YOLOv5+ECANet has the best recalls for RS, PS, PA and SC. This shows that the miss-selection rates of the improved algorithms are lower than the original YOLOv5 algorithm.

YOLOv4 algorithm and Faster R-CNN algorithm are far lower than the improved YOLOv5 algorithms in the fields of the mean average precision and the average precisions of all types of defects. In terms of F1-score and recall, YOLOv5+CBAM is higher than YOLOv4 algorithm, which indicates that YOLOv5+CBAM is better than YOLOv4 algorithm to a certain extent and also obtains the lower missed detection rates. Although Faster R-CNN algorithm has the best recalls for all types of defects, its precisions for all types of defects are very low, resulting in the higher false-selection rates of Faster R-CNN algorithm. It is undesirable to reduce the miss-selection rates by increasing the false-selection rates. The F1-scores of Faster R-CNN algorithm for all types of defects are far lower due to the polarization of the precisions and the recalls, so the Faster R-CNN algorithm is worse than the YOLOv5+CBAM algorithm to some extent. The average precision of Faster R-CNN algorithm for RS defects with a large number of small targets is low, and, therefore, Faster R-CNN algorithm has poorer detection ability for small targets.

Overall, the detection ability of the improved algorithms in this paper are better than the original YOLOv5 algorithm. Among them, YOLOv5+CBAM performs best in the mean average precision, its mean precisions for RS, PA and IN with a large number of small targets are increased by 8.1%, 6.58% and 4.01%, respectively, and the mean precisions for PS and SC with more extreme aspect ratio targets are increased by 0.93 and 3.26, respectively. Although YOLOv5+CBAM increases the average precision of PA, which is a small increase, both YOLOv5+SENet and YOLOv5+ECANet only decrease the average precision of PA. The improved algorithms do not have a large increase in the average precision of PA compared with the average precisions of the other four defects. This may be because the features of PA defects are more obvious, and the original YOLOv5 algorithm can extract a large number of effective features of targets of PA defects, and its average precision for PA is as high as 93.23%, but after adding the attention mechanism, the algorithm extracts a large number of features of non-target regions, so it leads to the improved algorithms not having a large increase in the average precision for PA, which also indicates that the ability of the attention mechanism is limited.

The results of YOLOv5+CBAM compared with YOLOv3~5 algorithms and Faster R-CNN algorithm in terms of detection speed are shown in Table 3.

**Table 3.** Comparison of detection speeds of YOLO algorithms, Faster R-CNN algorithm and YOLOv5+CBAM (in FPS).

| YOLOv3 | YOLOv4 | YOLOv5 | Faster R-CNN | YOLOv5+CBAM |
|---|---|---|---|---|
| 34 | 40 | 52 | 12 | 45 |

In Table 3, YOLOv5+CBAM is much faster than Faster R-CNN algorithm and YOLOv3~4 algorithms and slightly slower than the YOLOv5 algorithm, but the mean average precision of the YOLOv5 algorithm is lower. Therefore, in consideration of detection accuracy and detection speed, YOLOv5+CBAM is the optimal method.

In Figure 9, five types of defects in the dataset are randomly selected and marked with green rectangular boxes in this paper. The YOLOv5 algorithm and YOLOv5+CBAM are used to detect these five images, respectively, and the prediction results are compared with the ground truth boxes of the images.

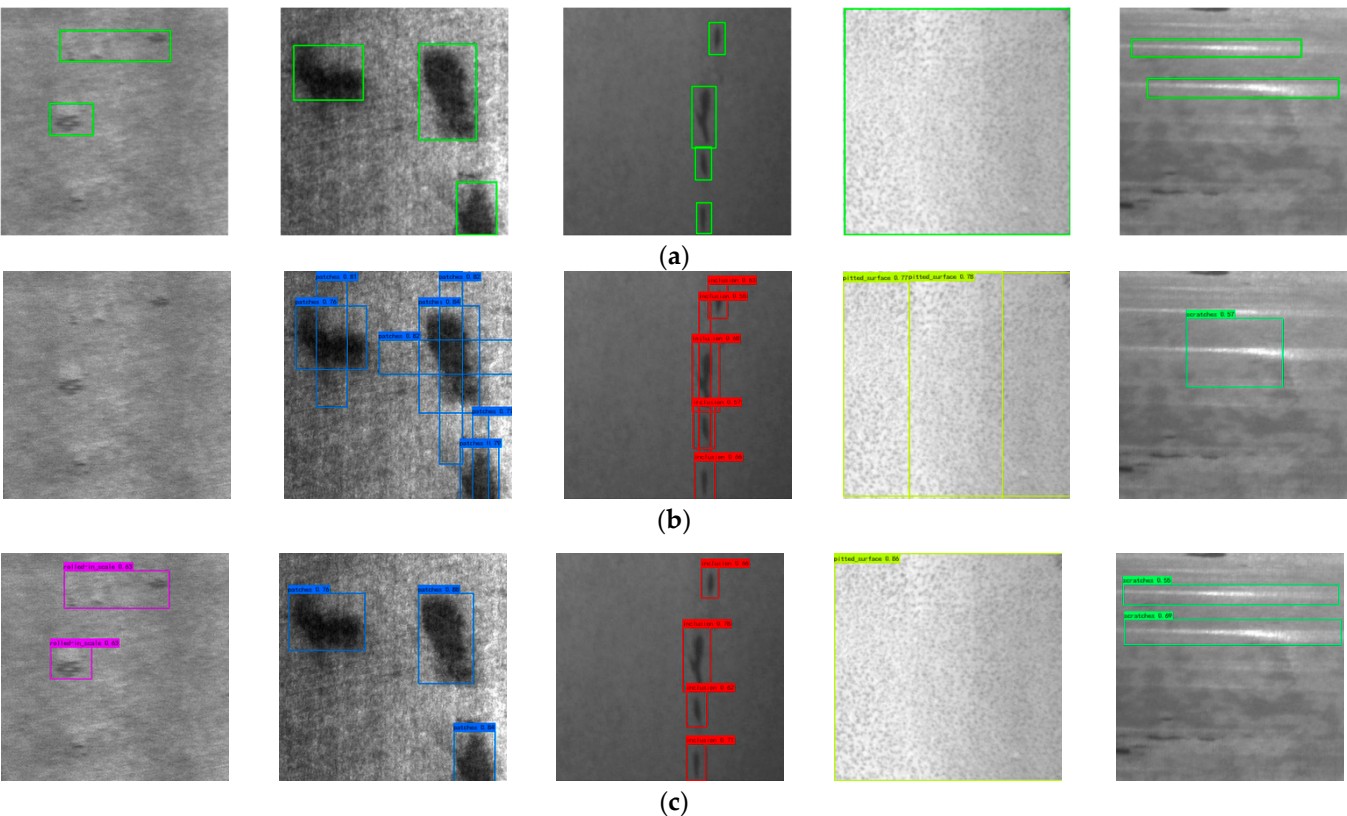

**Figure 9.** The prediction results of algorithms on RS, PA, IN, PS and SC. (**a**) Ground truth boxes for the five types of defects in the dataset. (**b**) Prediction boxes for the five types of defects using the YOLOv5 algorithm. (**c**) Prediction boxes for the five types of defects using YOLOv5+CBAM.

RS defects contain a large number of small targets and are not obvious in their features. Although the YOLOv5 algorithm fails to detect them, YOLOv5+CBAM has successful detections and predicts them very accurately. PA defects and IN defects contain a large number of small targets and extreme aspect ratio targets, which are obvious in their features. Both YOLOv5 algorithm and YOLOv5+CBAM have successful detections of them, but the predictions of YOLOv5+CBAM are more accurate compared with the ground truth boxes, but the prediction boxes of YOLOv5 algorithm contain a large amount of non-target information. SC defects and PS defects contain a large number of extreme aspect ratio targets, which are more accurately predicted by YOLOv5+CBAM, compared to ground truth boxes.

## 4. Conclusions

In this paper, YOLOv5 algorithm is improved according to the size characteristics of steel surface defects. The algorithm ability to detect small targets is improved by adding an attention mechanism to enhance the attention of the algorithm to the information of the small targets. The clustering analysis of the marked steel surface defect dataset is performed by the K-means algorithm, which enhances the algorithm ability to detect extreme aspect ratio targets. Experiments show that the mean average precision of the improved algorithm in this paper is higher than that of the original YOLOv5 algorithm. In particular, the CBAM attention mechanism has the best performance. The YOLOv5+CBAM algorithm has a significant increase in the average precisions for all types of defects. In the actual prediction

of defects, YOLOv5+CBAM outperforms the original YOLOv5 algorithm for small targets and extreme aspect ratio targets. Overall, the improvements of YOLOv5 algorithm by attention mechanism and K-means algorithm in this paper are effective for improving the detection of small targets and extreme aspect ratio targets, thus, improving the overall detection ability of the algorithm for steel surface defects.

This paper focuses on the task of steel surface defect detection and improves the original YOLOv5 algorithm for a large number of small targets and extreme aspect ratio targets of steel surface defects. Although the recognition accuracy has been increased, there are still some problems:

1.  The images of steel surface defects used in this paper are carefully selected and differ from those in the real production process. The algorithm needs to be further tested and adjusted in real-world scenes with real results.
2.  In the defect detection tasks, avoiding missed detection is more important than avoiding false detection. Therefore, during the training of the algorithm, it is important to pay attention not only to the average precisions of all types of defects and the mean average precision but also to the recalls of all types of defects, to improve the algorithm according to the changes of the recalls.
3.  The number of parameters of the algorithm is increased due to the addition of one attention mechanism to the algorithm. The follow-up work focuses on the research and implementation of the lightweight algorithm.

**Author Contributions:** Conceptualization, J.S.; formal analysis, J.S.; investigation, J.S.; writing—original draft preparation, J.Y.; writing—review and editing, J.Y.; supervision, Y.Z. All authors have read and agreed to the published version of the manuscript.

**Funding:** This paper was supported by the Fundamental Research Funds for the Local Provincial Universities of Hei longjiang Province in 2018 (Grant No. 2018-KYYWF-1189).

**Data Availability Statement:** Publicly available datasets were analyzed in this study. This data can be found here: https://aistudio.baidu.com/aistudio/datasetdetail/18746.

**Acknowledgments:** We would like to acknowledge the reviewers for their careful reading, helpful comments, and constructive suggestions, which have significantly improved the presentation of our manuscript.

**Conflicts of Interest:** The authors declare no conflict of interest.

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
