# Peer review of "Research on Steel Surface Defect Detection Based on YOLOv5 with Attention Mechanism"

_electronics, doi:10.3390/electronics11223735_

Round 1

Reviewer 1 Report

This manuscript presents an improved algorithm for defect detection on steel surfaces. Although this subject is interesting, the authors should perform various mandatory changes to their manuscript.

In the Abstract section, lines 10-13 the authors should be more specific about the problems "in the extraction and production process", the types of defects on the steel surfaces and the target detection tasks.

In the Introduction section, the authors should describe in more details the results of the previous works in the relevant literature regarding the types of defects detected, their size, the extent of success of the various algorithms used. Moreover, they should describe the advantages and disadvantages of previously used algorithms in detail.

The fourth and fifth paragraph of the Introduction section are too general. The authors should justify their statements according to the relevant literature in a more concise way. The statements about the "domestic" situation and the situation in "foreign countries" should be removed from the Introduction section.

The authors should justify their novelty based on state-of-the-art works.

In section 2, the authors should justify the relevance of the dataset to the required task (defect detection in steel specimens). The division of the dataset into training, validation and test set should be explained and further justified as the percentage of training sets is rather high and could lead to overfitting.

More details about each type of defect mentioned in page 3 should be provided.

In the conclusions section the authors should present more specific conclusions based on the comparison of different attention mechanisms and algorithms.

Author Response

Point 1: In the Abstract section, lines 10-13 the authors should be more specific about the problems "in the extraction and production process", the types of defects on the steel surfaces and the target detection tasks.

Response 1: We provided specific explanations of “the problems” and “all types of defects”. “the problems” were “the irresistible factors of material properties and processing technology”. “all types of defects” were “rolling scale, patches and so on”.

Point 2: In the Introduction section, the authors should describe in more details the results of the previous works in the relevant literature regarding the types of defects detected, their size, the extent of success of the various algorithms used. Moreover, they should describe the advantages and disadvantages of previously used algorithms in detail.

Response 2: We described the improvement work done by the authors for the small targets in the previous work, the experimental results, and the advantages and disadvantages of the algorithm in section 1.

Point 3: The fourth and fifth paragraph of the Introduction section are too general. The authors should justify their statements according to the relevant literature in a more concise way. The statements about the "domestic" situation and the situation in "foreign countries" should be removed from the Introduction section.

Response 3: We revised the section 1 to provide a more detailed description of the relevant literature and the work done in this paper in a more concise way. And we removed the statement about the “domestic” situation and the situation in "foreign countries".

Point 4: The authors should justify their novelty based on state-of-the-art works.

Response 4: We selected the literature using deep learning techniques for steel surface defect detection are from the last three years, and they were advanced enough to prove the thesis of our article.

Point 5: In section 2, the authors should justify the relevance of the dataset to the required task (defect detection in steel specimens). The division of the dataset into training, validation and test set should be explained and further justified as the percentage of training sets is rather high and could lead to overfitting.

Response 5: In section 2.1, we described the dataset in more detail and provided a full description of the division of the training, validation and test sets. Overfitting usually causes the training loss to drop suddenly and substantially, while the validation set loss starts to rise. We set the epoch of training to 300, the validation set loss was always slightly higher than the training set loss during the training process, and they almost keep decreasing. So we consider that the algorithm is not overfitted.

Point 6: More details about each type of defect mentioned in page 3 should be provided.

Response 6: In the third to eleventh paragraph of section 2.1, we provided a more detailed description about each type of defect, including their size and appearance.

Reviewer 2 Report

The study incorporated three types of attention modules to YOLOv5 respectively, tested the efficacy of the improved YOLOv5 using steel surface images with small and large detects, and showed that all the attention modules were able to improve YOLOv5 overall performance. Such a result would be worth reporting, if, the proposed methodology is novel. However, there are many major issues with the current manuscript. A significant revision is needed before I can properly evaluate the manuscript, hence I do not consider this manuscript as complete and suitable for publication. Below are my specific comments.

• Inappropriate grammar or language is an outstanding issue of the manuscript. I will only use abstract and introduction as an example and will no longer comment on any further grammar mistakes, which exits throughout the manuscript. The authors need to work with native English speakers or utilize English editing service from the editorial office to thoroughly check the language of the manuscript.

o Line 10, “some problems”, “some defects”, what are they? Be specific.

o Line 13, “so the detection accuracy is low” makes no sense because no context is given. What algorithm? How low of an accuracy is low?

o Line 13-14, “the attention mechanism is” should be “attention mechanisms were”.

o Line 14, “origin” should be “original”.

o Line 17, “show” should be “showed”.

o Line 18, “are” should be “were”.

o Line 19, “algorithm” should be “algorithms”

o Line 24, your keywords should not repeat title.

o Line 35, 56 “at home and abord” unnecessary.

o Line 37, 39, acronyms should go into parentheses.

o Line 70, inappropriate to use “domestic” and “foreign countries” since Electronics is an international journal.

• The study is not properly justified due to weak literature review. The authors need to first review existing studies using object detectors for steel defect detection, and then review existing studies improving CNN networks using attention modules (line 46-55 lack a clear focus). A comprehensive literature review is needed.

• What is the knowledge gap in current literature? What is the necessity of the study? How is your study different from and superior than [13] (line 53-55). Again, a comprehensive literature review is needed to cover similar studies.

• Clearly define your objectives in introduction.

• Clearly define RS, PA, IN, PS, SC.

• Provide description and explanation in figure 1. The image annotations are not intuitive and some of them do not make sense. For example, figure 1a makes no sense to me, why figure 1b has 2 boxes instead of just 1, the 2 boxes in figure 1d contain very different looking defects, why figure 1e does not have at least 3 boxes since there is clearly another scratch to the left of the boxes.

• Very little information is share regarding the dataset. How are image collected? What is your steel sample size? Where did your defected steel sample come from? How are image annotation prepared? Who prepared? What are the image annotation rules? How is image annotation quality evaluated before model training?

• Line 106-122, define small/large targets.

• Line 103-128, unnecessary. Why bother using width and height to evaluate target size? Shouldn’t you directly use area (width*height) to evaluate target size?

• Equation 3, I doubt this is the correct definition. Usually it is 11-point precision-recall curve interpolation under a certain IoU.

• What is your confidence and IoU cutoffs for precision and recall?

• Results are not analyzed and discussed in detail. For example, table 1, the attention modules lacked a consistency in terms of how they can improve YOLOv5 accuracy. SENet mostly improved RS, CBAM mostly improved RS and PS, ECANet mostly improved PS, IN and SC. Meanwhile the accuracy for the rest classes either stayed roughly the same or even decreased. When you claim the attention modules can improve accuracy for small targets, how come each of them only work for certain classes? How does one attention module work better for certain classes than the other two modules?

• What is the value and necessity of figure 8?

• If comparing your improved YOLOv5 to YOLOv4 and Faster R-CNN is always one of your objectives then YOLOv4 and Faster R-CNN should be introduced in introduction.

• Table 1 and 2 can be combined.

• Why figure 1 and 9 are using the same set of images? Do you have a different set of images to present?

• Line 321, how are the images carefully selected? Why nothing is mentioned in the main text of the article?

Author Response

Point 1: Inappropriate grammar or language is an outstanding issue of the manuscript. I will only use abstract and introduction as an example and will no longer comment on any further grammar mistakes, which exits throughout the manuscript. The authors need to work with native English speakers or utilize English editing service from the editorial office to thoroughly check the language of the manuscript.

Response 1: Firstly, we modified and optimized the article using the appropriate grammar. Secondly, we modified the keywords of the article, and removed the statement about the “domestic” situation and the situation in "foreign countries".

Point 2: The study is not properly justified due to weak literature review. The authors need to first review existing studies using object detectors for steel defect detection, and then review existing studies improving CNN networks using attention modules (line 46-55 lack a clear focus). A comprehensive literature review is needed.

Response 2: Firstly, we described early defect detection algorithms, which mainly used digital image processing techniques and machine learning techniques, and then evaluated these algorithms. Next, we described three previous works that improve deep learning algorithms for small targets, one of which used the attention mechanism, and described the advantages and disadvantages of these improved algorithms.

Point 3: What is the knowledge gap in current literature? What is the necessity of the study? How is your study different from and superior than [13] (line 53-55). Again, a comprehensive literature review is needed to cover similar studies.

Response 3: We described the improvement work done by the authors for the small targets in the previous work, the experimental results, and the advantages and disadvantages of the algorithms in the Introduction section.

Point 4: Clearly define your objectives in introduction.

Response 4: In the fourth to eleventh paragraph of section Introduction, we defined our objects. Firstly, for steel surface defects with a large number of small targets, an attention mechanism was used to enhance the attention of the algorithm to the regions of the small targets. Secondly, for the extreme aspect ratio targets, the K-means algorithm was used to cluster the anchor boxes, so that the anchor boxes can be adapted to all types of sizes, especially for extreme aspect ratio targets. The experiments showed that the improved algorithm in this paper can improve the detection abilities of small and extreme aspect ratio targets, which can provide a reference for the automatic detection of steel defects.

Point 5: Clearly define RS, PA, IN, PS, SC.

Response 5: In the second paragraph of section 2.1, the names of defect types in the dataset were abbreviated, i.e. RS is for rolled-in_scale, PA is for patches, IN is for inclusion, PS is for pitted_surface, and SC is for scratches,in order to further facilitate the proper names.

Point 6: Provide description and explanation in figure 1. The image annotations are not intuitive and some of them do not make sense. For example, figure 1a makes no sense to me, why figure 1b has 2 boxes instead of just 1, the 2 boxes in figure 1d contain very different looking defects, why figure 1e does not have at least 3 boxes since there is clearly another scratch to the left of the boxes.

Response 6: We provided description and explanation in figure 1. According to the image labels of the dataset, we used the green rectangle boxes to mark the locations of the defects.

Point 7: Very little information is share regarding the dataset. How are image collected? What is your steel sample size? Where did your defected steel sample come from? How are image annotation prepared? Who prepared? What are the image annotation rules? How is image annotation quality evaluated before model training?

Response 7: We selected images of steel surface defects from the Northeastern University steel surface defect dataset (NEU-CLS). NEU-CLS is a general dataset with high quality.

Point 8: Line 106-122, define small/large targets.

Response 8: In the fourth paragraph of section 2.1, we provided definitions for small targets and extreme aspect ratio targets. “Targets whose width and height both belong to less than 80 pixels are defined as small targets. Among the two values of width and height of a target, one of which is larger than 150 pixels and the other is smaller than 100 pixels, the target is defined as an extreme aspect ratio target”.

Point 9: Line 103-128, unnecessary. Why bother using width and height to evaluate target size? Shouldn’t you directly use area (width*height) to evaluate target size?

Response 9: We think that small targets cannot be defined based on area, because many extreme aspect ratio targets have equally small areas, but it is not appropriate to define them as small targets. So we analyzed the sizes of all types of defects using scatter plots, then defined small targets and extreme aspect ratio targets, and improved the algorithm for small targets and extreme aspect ratio targets.

Point 10: Equation 3, I doubt this is the correct definition. Usually it is 11-point precision-recall curve interpolation under a certain IoU.

Response 10: In the 6th paragraph of section 2.3, we modified equation 3 and provided the correct equation for the average precision. 

Point 11: What is your confidence and IoU cutoffs for precision and recall?

Response 11: In the second and third paragraphs of section 2.3, we set the confidence and IoU of the algorithm to 0.5 and 0.3, respectively.

Point 12: Results are not analyzed and discussed in detail. For example, table 1, the attention modules lacked a consistency in terms of how they can improve YOLOv5 accuracy. SENet mostly improved RS, CBAM mostly improved RS and PS, ECANet mostly improved PS, IN and SC. Meanwhile the accuracy for the rest classes either stayed roughly the same or even decreased. When you claim the attention modules can improve accuracy for small targets, how come each of them only work for certain classes? How does one attention module work better for certain classes than the other two modules?

Response 12: In section 3.3, we modified the improvement of the algorithm by only adding an attention mechanism module in the process of passing the shallow feature map from the Backbone structure to the FPN structure, with the aim of improving the algorithm's attention to the shallow feature map as a way to improve the algorithm's ability to detect small targets. And the anchor boxes were clustered and analyzed using K-means algorithm as a way to improve the algorithm ability to detect extreme aspect ratio targets so that the anchor boxes can be adapted to all types of sizes, especially for extreme aspect ratio defects. The mean average precisions of all three improved algorithms were improved. Both YOLOv5+SENet and YOLOv5+ECANet only decreased the average precision of PA, although YOLOv5+CBAM increased the average precision of PA, which was a small increase. Overall, three types of attention mechanisms had a consistency in terms of improving YOLOv5 accuracy.

Point 13: What is the value and necessity of figure 8?

Response 13: We removed figure 8.

Point 14: If comparing your improved YOLOv5 to YOLOv4 and Faster R-CNN is always one of your objectives then YOLOv4 and Faster R-CNN should be introduced in introduction.

Response 14: In the third paragraph of section 1, we introduced the YOLOv4 algorithm and the Faster RCNN algorithm.

Point 15: Table 1 and 2 can be combined.

Response 15: In page 12, we combined Table 1 and 2.

Point 16: Why figure 1 and 9 are using the same set of images? Do you have a different set of images to present?

Response 16: In figure 1, we initially showed the appearance of all types of defects. Figure 9 showed the actual results of defect detection using our algorithm, and then the results were compared and analyzed. The same images were used in figure 1 and figure 9 in order to have consistency between section 2 and section 3 of the article. According to your suggestions, we replaced the set of images in Figure 9 to make the conclusion more convincing.

Point 17: Line 321, how are the images carefully selected? Why nothing is mentioned in the main text of the article?

Response 17: In the third paragraph on page 9, five images in the dataset were randomly selected and marked with green rectangular boxes in this paper. 

Reviewer 3 Report

My correction with the manuscript are relatively minor, and are listed below.

Corrections are needed in the presentation of the results, both in the standardization of font size, image resolutions and in the naming of images. Indeed, some images are impossible to interpret because of the titles they bear, especially because of the numerous abbreviations that have not been explained.

Author Response

Point 1: Corrections are needed in the presentation of the results, both in the standardization of font size, image resolutions and in the naming of images. Indeed, some images are impossible to interpret because of the titles they bear, especially because of the numerous abbreviations that have not been explained.

Response 1: We revised font size, image resolutions and in the naming of images according to the standard.

Round 2

Reviewer 1 Report

The authors performed some of the requested modifications to their manuscript. However, it is still required that they conducted a more comprehensive literature review in the Introduction section regarding previous works (at least 10), by presenting specific details such as the types of defects examined in these works, their size and reported problems in their detection.

Author Response

Point 1: The authors performed some of the requested modifications to their manuscript. However, it is still required that they conducted a more comprehensive literature review in the Introduction section regarding previous works (at least 10), by presenting specific details such as the types of defects examined in these works, their size and reported problems in their detection.

Response 1: In the second to third paragraph of section Introduction, we referred to the literature of ten related studies, the first four of which were traditional methods and the last six were deep learning methods. For the first four literatures, we provided method descriptions and analysis of strengths and weaknesses for them as a whole. For the last six literatures, we provided detailed method descriptions and analyses of strengths and weaknesses for each of them, and we provided detailed the types of defects and the sizes of defects for the literature that used a different dataset than ours.

Reviewer 2 Report

The authors have sufficiently addressed most of my comments. However, there is one point that I am not satisfied with. I previously asked, “how is your study different from and superior than [13] (line 53-55)”, which is about a study (Zeqiang, S., & Bingcai, C. (2022). Improved Yolov5 Algorithm for Surface Defect Detection of Strip Steel. In Artificial Intelligence 367 in China (pp. 448-456). Springer, Singapore.) that seems to be rather relevant and similar to the current study. Not only my question was not answered, but the citation of the study was also removed in the revised manuscript. I would ask the authors to add the citation back and properly address my comment in the manuscript. Specifically, what is the novelty of the current study compared to study [13]? I will recommend major revision so that I can have the opportunity to check whether this point is addressed in the next revised manuscript.

Author Response

Point 1: The authors have sufficiently addressed most of my comments. However, there is one point that I am not satisfied with. I previously asked, “how is your study different from and superior than [13] (line 53-55)”, which is about a study (Zeqiang, S., & Bingcai, C. (2022). Improved Yolov5 Algorithm for Surface Defect Detection of Strip Steel. In Artificial Intelligence 367 in China (pp. 448-456). Springer, Singapore.) that seems to be rather relevant and similar to the current study. Not only my question was not answered, but the citation of the study was also removed in the revised manuscript. I would ask the authors to add the citation back and properly address my comment in the manuscript. Specifically, what is the novelty of the current study compared to study [13]? I will recommend major revision so that I can have the opportunity to check whether this point is addressed in the next revised manuscript.

Response 1: According to your comments, we added the citation back, i.e. [22]. Firstly, the type and location of the attention mechanism used was different from ours. The literature [22] added an attention mechanism after the pooling structure of the FPN network, which we considered as an untargeted approach. And pooling led to a large amount of feature information loss, resulting in the attention mechanism not getting enough feature information. But we added an attention mechanism was targeted to small targets. Secondly, the literature [22] made other improvements to the algorithm. Compared to our improved algorithm, although the improvement of the YOLOv5 algorithm in the literature [22] was huge, the increase of the mean average precision was smaller and the detection speed decreased significantly.

Round 3

Reviewer 1 Report

The authors performed the necessary changes to their manuscript, thus it can be considered for publication.

Author Response

Dear Reviewer, Thank you for your comments.

Reviewer 2 Report

The study “Zeqiang, Sun, and Chen Bingcai. "Improved Yolov5 Algorithm for Surface Defect Detection of Strip Steel." Artificial Intelligence in China. Springer, Singapore, 2022. 448-456.” is rather similar to the current study in all aspects. Both studies used the same dataset and tested Faster-RCNN and YOLOv5. While the mAPs of YOLOv5 in the two studies are 82.0% and 81.78% respectively, which are similar enough and make sense, how come the mAPs of Faster-RCNN are 75.14 % and 31.96% respectively and differ so greatly? Why was this point not brought up in the manuscript?

Author Response

Point 1: The study “Zeqiang, Sun, and Chen Bingcai. "Improved Yolov5 Algorithm for Surface Defect Detection of Strip Steel." Artificial Intelligence in China. Springer, Singapore, 2022. 448-456.” is rather similar to the current study in all aspects. Both studies used the same dataset and tested Faster-RCNN and YOLOv5. While the mAPs of YOLOv5 in the two studies are 82.0% and 81.78% respectively, which are similar enough and make sense, how come the mAPs of Faster-RCNN are 75.14 % and 31.96% respectively and differ so greatly? Why was this point not brought up in the manuscript?

Response 1: During several previous training processes using the Faster-RCNN algorithm, in order to reflect the realistic comparison effect, we used the default anchor_size of the original Faster R-CNN algorithm, which was obtained from the VOC dataset. VOC dataset was a general dataset, most of its images came from real-life scenes. The sizes of the ground truth boxes of VOC dataset were quite different from the sizes of images of our dataset. Due to our dataset included a large number of small targets, we modified the anchor_size to be suitable for a large number of small targets in our dataset, then trained and tested this algorithm. Finally, the mean average precision of this algorithm was 74.85%. The experiments showed that the default anchor_size was not suitable for our dataset, so the mean average precision was low. By modifying the anchor_size, the mean average precision was increased substantially. In Table 2, we modified the data of Faster R-CNN. In the fourth paragraph below Table 2, we modified the comparative analysis of the algorithms.